# Outcome Expectations on Physical Activity: The Roles of Body Appreciation and Health Status

**DOI:** 10.3390/bs15030394

**Published:** 2025-03-20

**Authors:** Nanbo Wang, Qingli Guan, Zihan Yin, Song Zhou, Wenbo Zhou

**Affiliations:** 1Department of Psychology, School of Health, Fujian Medical University, Fuzhou 350122, China; wangnb@fjmu.edu.cn; 2School of Psychology, Fujian Normal University, Fuzhou 350117, China; qsx20220445@student.fjnu.edu.cn (Q.G.); zhousong@fjnu.edu.cn (S.Z.); 3Dalian Leicester Institute, Dalian University of Technology, Dalian 124221, China; 20223292094@mail.dlut.edu.cn; 4China Basketball College, Beijing Sport University, Beijing 100084, China

**Keywords:** outcome expectations, physical activity, body appreciation, health status

## Abstract

Although the research has demonstrated that outcome expectations influence physical activity, the precise underlying mechanisms remain ambiguous. Therefore, this study employed a cross-sectional research design to investigate the effect of outcome expectations on physical activity and to examine the roles of body appreciation and health status. A total of 1349 participants (*M*_age_ = 19.42, *SD* = 1.51; 410 males, 939 females) were recruited for this study and the data were analyzed using a combination of path analysis and network analysis. The results reveal that outcome expectations positively predict physical activity. Body appreciation mediated the relationship between outcome expectation and physical activity. Furthermore, health status moderated the relationship between body appreciation and physical activity. Specifically, body appreciation did not influence physical activity for individuals with a low health status. Network analysis further revealed that there were more edges between body appreciation and physical activity in the high health status group compared to the low health status group. For individuals with a high health status, BA1 and PA1 exhibited the strongest connection among all the edges between body appreciation and physical activity. Outcome expectations play a significant role in physical activity, with body appreciation acting as a mediator. Health status moderates the effect of body appreciation on physical activity, suggesting that interventions targeting body appreciation may be more effective for individuals with a better health status. These findings offer insights for tailored physical activity interventions.

## 1. Introduction

Adequate physical activity benefits individuals’ physical and mental health, including optimizing cardiovascular function, managing stress, and improving psychological well-being ([18]; [23]). Nevertheless, a significant decline in physical activity among university students was prevalent ([62]). Less than 50% of college students acknowledged meeting the current guideline recommendations for aerobic exercise levels ([2]). In China, participation in physical activity among university students is similarly low. For instance, a recent study involving 41,620 Chinese university students found that only 30.2% engaged in moderate-to-vigorous physical activity ([31]). Meanwhile, national and social concerns regarding the health status of Chinese university students are constantly escalating. Consequently, it is necessary to motivate university students to engage in physical activity by implementing psychological strategies. Drawing upon the previous literature, it is evident that outcome expectations have the potential to promote participation in physical activity ([29]; [30]). However, the mechanisms underlying this relationship remain unclear. In order to explore the relationship between outcome expectations and physical activity, this study was conducted to comprehend this process from the perspectives of body appreciation and health status.

### 1.1. Outcome Expectation and Physical Activity

Outcome expectation, or outcome expectancy, refers to an individual’s beliefs regarding the positive and negative consequences anticipated to arise from engaging in a specific behavior ([8]; [39]). It is a pivotal construct within Bandura’s social cognitive theory, Pender’s health promotion model, and Schwarzer’s Health Action Process Approach ([8]; [39]; [49]). Outcome expectations comprised three interrelated yet conceptually distinct sub-domains: physical (pertaining to body structure and function), social (related to interpersonal relationships), and self-evaluation (involving emotional and affective reflections) ([8], [9], [10]).

Numerous studies have revealed that positive alterations in outcome expectations can exert a facilitative impact on physical activity ([29]; [30]; [41]; [51]). For instance, [30] ([30]) found that outcome expectations not only directly promote physical activity, but also mediate the relationship between other psychological factors (e.g., self-efficacy and decisional balance) and physical activity. This may be related to the expectations and attitudes people hold about the outcome of their behavior. According to the social cognitive theory, when individuals perceive that specific behavior will lead to positive outcomes that are significant to them, the likelihood of engaging in that behavior increases ([8]). Positive outcome expectations (e.g., anticipation of enhanced health status) can serve as motivational stimuli for individuals to engage in daily physical activity. In contrast, negative outcome expectations (e.g., expectation of heightened pain and fatigue) might induce discomfort in daily health behaviors, which may lead individuals to give up such behaviors ([8]). In fact, individuals are more inclined to believe that physical activity can enhance positive expectations regarding bodily outcomes than to perceive it as a means of reducing negative expectations ([29]). Those who anticipate favorable health outcomes tend to report longer durations of physical activity compared to individuals with neutral or negative health outcome expectations ([26]). Furthermore, individuals in better physical conditions or those with fewer disabilities tend to have higher expectations regarding physical activity, rather than increased expectations concerning social or self-evaluation outcomes ([61]). Given these insights, our study first aims to establish whether outcome expectations positively predict physical activity, which serves as the foundation for exploring potential mediators and moderators in subsequent analyses. Therefore, we propose the following hypothesis:

**H1:** 
*Outcome expectations positively predict physical activity.*


### 1.2. The Mediating Role of Body Appreciation

Body appreciation serves as a symbol of a positive body image, encompassing the acceptance of one’s body, cultivating body-affirming attitudes, and honoring the body by meeting its needs and engaging in health-promoting behaviors ([6]; [56]). As proposed by the expectancy theory, an individual’s motivation to engage in a particular behavior is influenced by their confidence in the expected outcomes of their efforts (expectancy), their belief that specific actions will lead to certain performances (instrumentality), and the importance they attribute to those outcomes (value) ([58]). In other words, when individuals possess high expectations of positive results—believing that physical activity will yield beneficial physical and health outcomes—they are likely to develop a greater appreciation for their bodies, as they recognize the potential and value of their physical improvements. Similarly, people may have the expectation that physical activity can help shape a good body and mental outlook, which in turn influences perceptions of the body and the actions actually taken ([63]). Positive expectations about taking action can motivate more positive attitudes and behaviors.

Conversely, negative expectations reduced the likelihood of positive behaviors occurring, as well as more negative attitudes ([8]; [30]). For example, a longitudinal study of weight loss expectations and body image revealed that when the discrepancy between weight loss expectations and future actual weight outcomes was large, it led to more dissatisfaction with body image and more worry about weight ([14]). This indicated that unrealistic outcome expectations can exacerbate a negative body image (such as body image dissatisfaction) and diminish the acceptance of one’s body, thereby impacting body appreciation. However, when individuals have more realistic and positive outcome expectations, it may facilitate an acceptance and appreciation of their bodies, enhancing their overall body appreciation. Thus, positive outcome expectations may contribute to an increased level of body appreciation. As proposed by the Health Action Process Approach (HAPA) theory, individuals’ expectations of behavioral outcomes predict the formation of their behavioral intentions and the actual behaviors they adopt ([47]; [49]).

Considerable research has emphasized the constructive role of body appreciation in fostering engagement in health behaviors, particularly physical activity ([5]; [15]; [37]; [42]). For example, a longitudinal study in adolescents showed that individuals with higher levels of body appreciation were associated with higher rates of physical activity participation over a one-year observation period ([4]). Conversely, a lack of appreciation for one’s body was associated with unhealthy habits, including diminished physical activity ([11]; [54]). This positive association may be attributable to the reinforcement of intrinsic motivation for physical activity facilitated by body appreciation ([15]). When individuals have respect and honor for their bodies, they demonstrate a greater tendency to cultivate them ([7]). Indeed, exercise itself can foster an acceptance and appreciation of one’s body, thereby fostering confidence in one’s current physical state ([36]). For example, the exercise intervention group experienced improvements in body appreciation, body connectedness, and body satisfaction compared to the non-intervention group ([24]). In light of these ideas, we propose the following hypothesis:

**H2:** 
*Body appreciation mediates the relationship between outcome expectation and physical activity.*


### 1.3. The Moderating Role of Health Status

Health has conventionally been defined as the absence of a disease or illness ([46]). However, contemporary perspectives tend to conceptualize health as a subjective construct rooted in individual life experiences, shaped by individual feelings and behaviors, even in the context of physical illness ([32]). Indeed, the health research has increasingly emphasized perceived health and other subjective appraisals of health, with self-rated health emerging as a valid indicator of actual health status ([20]).

The relationship between body appreciation and physical activity may vary depending on health status. The research suggests that health status not only influences an individual’s perception of their body, but may also further affect their behavior through this perception ([40]). Individuals in good health are more likely to have positive bodily experiences and a favorable self-image, which fosters greater body appreciation and, consequently, a greater tendency to engage in physical activity ([21]). These individuals may experience optimal physical functioning, enabling them to more flexibly perform daily activities or exercise to maintain their health ([59]). Those who are more inclined to take proactive measures to preserve their health often display a better posture and physical condition, accompanied by a positive attitude toward their bodies ([34]). Conversely, individuals with poor health may develop negative perceptions of their body image and functionality due to declines in physical capacity or the effects of illness ([3]). Such negative body perceptions often diminish their motivation to engage in physical activity ([52]). For instance, studies have found that individuals with a disease or physical limitations may have negative views of their body image and subsequently reduce health-promoting behaviors in their daily lives ([3]). Furthermore, mental health issues, such as depression and anxiety, may weaken individuals’ positive body evaluations, further hindering their participation in physical activity ([52]). Based on the above concepts, we propose the following hypothesis:

**H3:** 
*Health status moderates the relationship between body appreciation and physical activity.*


In summary, the present study aims to develop a moderated mediation model to examine the mechanism by which outcome expectations influence physical activity (Figure 1). The network analysis is used to explore the relationship between outcome expectations, body appreciation, and physical activity at the item dimension for the total sample and different health status levels to further provide support for the results of the structural equation model.

## 2. Method

### 2.1. Participants and Research Design

A total of 1387 participants were recruited for this study using a Chinese online platform (https://www.wjx.cn/). Inclusion criteria were: (1) Age ≥ 18 years; (2) No history of mental illness and normal cognitive ability; (3) No severe physical disabilities or diseases that affect daily activities; and (4) Voluntary participation and signed informed consent. Exclusion criteria were: (1) Age < 18 years; (2) History of mental illness or cognitive impairments, such as dementia; (3) Presence of disability or mobility limitations; and (4) Consecutive identical responses to all items on the questionnaire. After excluding 15 participants due to age < 18 years and 23 participants due to consecutive identical responses to all items on the questionnaire, the final valid sample comprised 1349 individuals (*M*_age_ = 19.42, *SD* = 1.51; 939 women), resulting in an effective response rate of 96.97% (Figure 2). Participants provided informed consent prior to enrollment in the study, which was approved by the Ethical Committee of Fujian Normal University (20210310), following the Declaration of Helsinki. Subsequently, they provided demographic information, including age, gender, and subjective social class. The subjective social class was measured using the MacArthur Scale of Subjective Social Status based on a 10-point scale ([1]). Additionally, they completed scales assessing outcome expectations, body appreciation, health status, and physical activity. The valid sample encompassed individuals with diverse educational backgrounds, comprising 42 (3.11%) in high school and below, 1 in junior college, 1294 (95.92%) undergraduates, and 12 (0.90%) postgraduates and beyond.

### 2.2. Measures

Outcome expectations were measured by three items adopted from [48]’s ([48]) research. The scale has been validated in Chinese populations ([65]), for example, “If I engage in physical activity, I can prevent disease”. Participants were asked to rate each item on a 5-point scale (1 = completely disagree; 5 = completely agree). A higher mean score indicated greater consideration of the behavioral outcomes. In the current study, the Cronbach’s alpha was 0.86.

Body appreciation was measured using the 10-item Body Appreciation Scale-2 ([55]), with the Chinese version of the scale demonstrating robust internal consistency ([53]). Participants were required to rate each item on a 5-point scale (1 = never, 5 = always). A higher mean score was indicative of greater levels of body appreciation. The Cronbach’s alpha was 0.95 in the current study.

Health status was measured by two items ([60]), “In general, how would you say your health is?”, where participants needed to score on a 5-point scale (1 = poor, 5 = excellent). Another was “Compared to other persons of my sex and age, my current health is?” Participants were still asked to use a 5-point scale (1 = much below average, 5 = much above average). A higher mean score indicated a better self-reported health status. The validity of this scale has been validated in the Chinese population ([64]). In the current study, the Cronbach’s alpha coefficient was 0.83.

Physical activity was assessed by two items adopted from a previous study ([48]). One of the items was: “In general, how often do you engage in physical activities per week on average?”. Participants were asked to rate this on a 5-point scale (1 = no exercise; 5 = 7 or more times). A higher mean score represents greater participation in a physical activity. The Chinese version of the scale has been validated ([50]). The Cronbach’s alpha was 0.82 in the current study.

### 2.3. Data Analysis

This study was conducted using R (version 4.3.0) and SPSS (version 26) for data analysis. Descriptive statistics, reliability analysis, and correlation analysis among variables were conducted in R, while the common method bias test and path analysis were executed using SPSS. Model 4 and Model 14 in the PROCESS macro program were used to analyze the mediating role of body appreciation and the moderating role of health status ([25]). Mediation effects were assessed by 5000 bootstraps to determine the range of confidence intervals. The data were standardized for model analysis. Additionally, network analysis was employed to examine the total sample, as well as high and low health status groups separately, in order to further illustrate potential differences. The Network Module in Jeffreys’s Amazing Statistics Program (Stathopoulou et al.) (version 0.18.3) was employed, utilizing the “EBICglasso” estimation ([19]). Each item was represented as a node in the network, with edges denoting connections between nodes. Within a network, certain nodes are dominant if they occupy central positions and exhibit stronger connections to other nodes ([12]). Non-parametric bootstrapping (1000 replicates) was conducted to estimate 95% confidence intervals of edge values. In the item-level network analysis, the abbreviation pattern follows the format: [construct abbreviation] + [item number]. For instance, OE1 and OE2 denote the first and second items assessing outcome expectations, respectively. The same convention applies to other constructs.

Four centrality indices (betweenness, closeness, strength, and expected influence) were computed for the nodes to provide an understanding of the importance of each node within the network structure ([38]). Betweenness of a node denotes the frequency of its involvement in connecting two other nodes along the shortest available path. Closeness signifies the inverse summation of the weights of the shortest paths from a given node to all other connected nodes, with higher values indicating susceptibility to influence from other nodes. Strength represents the cumulative weight of all edges directly linked to a node, with greater strength indicative of robust connections to other nodes ([17]). Higher expected influence reflects heightened sensitivity to change and the potential to serve as a hub by linking other pairs of nodes within the network ([27]). The bootnet package was employed to quantify correlation stability. A correlation stability value above 0.5 indicates greater stability. The NetworkComparisonTest package was utilized to investigate the differences in network structure and strength across varying levels of depression ([57]). The significance level was set at *p* < 0.05.

## 3. Results

### 3.1. Descriptive Statistics, Correlation Analysis, and Common Method Bias Test

The descriptive statistics for the variables, including means, standard deviations, and Pearson correlation coefficients, are presented in Table 1. The results of the correlation analysis show that that outcome expectation is positively correlated with body appreciation (*r* = 0.53, *p* < 0.001), health status (*r* = 0.39, *p* < 0.001), and physical activity (*r* = 0.20, *p* < 0.001). Body appreciation was positively related to health status (*r* = 0.56, *p* < 0.001) and physical activity (*r* = 0.20, *p* < 0.001). Health status was positively correlated with physical activity (*r* = 0.27, *p* < 0.001).

Confirmatory factor analysis (CFA) was employed to assess the severity of common method bias in the study, following the approach summarized by [43] ([43]). The fit indices for the four-factor model were *χ*^2^ = 1235.20, *df* = 113, *p* < 0.001, CFI = 0.94, TLI = 0.92, RMSEA = 0.09, and SRMR = 0.04. In contrast, when all items of the four variables were constrained to load onto a single factor, the fit indices were *χ*^2^ = 4144.72, *df* = 119, *p* < 0.001, CFI = 0.77, TLI = 0.74, RMSEA = 0.16, and SRMR = 0.09. Given that the fit indices of the four-factor model exhibited a superior performance compared to those of the one-factor model, it can be concluded that there was no serious common method bias in this study.

### 3.2. The Mediation Effect of Body Appreciation

We established a mediation model to examine the mediating role of body appreciation between outcome expectations and physical activity, controlling for gender, age, and subjective social class. The results, as presented in Table 2, show that outcome expectations have a significantly positive impact on physical activity, both for the total effect (*β* = 0.18, *SE* = 0.03, *p* < 0.001, 95%CI = [0.14, 0.23]) and the direct effect (*β* = 0.10, *SE* = 0.03, *p* < 0.001, 95%CI = [0.04, 0.16]). Thus, H1 is supported. Furthermore, outcome expectations positively predicted body appreciation (*β* = 0.50, *SE* = 0.02, *p* < 0.001, 95%CI = [0.46, 0.55]), while body appreciation positively predicted physical activity (*β* = 0.17, *SE* = 0.03, *p* < 0.001, 95%CI = [0.11, 0.22]). This indicated that body appreciation partially mediated the relationship between outcome expectations and physical activity, with a mediation effect of 0.08 (*SE* = 0.02, 95% CI = [0.04, 0.16]), accounting for 44.44% of the total effect. Thus, H2 is supported.

### 3.3. The Moderation Effects of Health Status

A moderated mediation analysis was conducted to examine the moderating role of health status in the relationships between body appreciation and physical activity, with variables centered. The results demonstrate a significant interaction coefficient between health status and body appreciation (*β* = 0.06, *SE* = 0.02, *p* = 0.002, 95%CI [0.02, 0.10]), suggesting a significant moderating effect of health status on the relationship between body appreciation and physical activity (see Table 3).

The simple slope analysis was performed to further examine the moderating role of health status. The result reveals that the positive effect of body appreciation on physical activity is significant among individuals with a high (M + 1 SD) health status (*β* = 0.15, *SE* = 0.04, *p* < 0.001, 95% CI [0.07, 0.23]), but not for those with a low (M − 1SD) health status (*β* = 0.03, *SE* = 0.04, *p* = 0.387, 95% CI [−0.04, 0.10]). As shown in Table 4, the indirect effect of outcome expectations on physical activity through body appreciation is stronger for the high-level health status (*β* = 0.08, *SE* = 0.02, 95% CI [0.03, 0.12]), whereas this indirect effect is not significant in the low-level health status (*β* = 0.02, *SE* = 0.02, 95% CI [−0.02, 0.06]). Thus, H3 is supported.

### 3.4. Network Structure

The network analysis was performed to explore the structure of outcome expectations, body appreciation, and physical activity at the item level for the total sample. The results reveal the network structure comprises 15 nodes and 72 non-zero edges (see Figure 3a and Appendix A, Table A1a). Among the within-variable edges, PA1 and PA2 exhibited the strongest connection (0.78). For inter-variable edges, OE1 and BA1 showed the strongest connection (0.16). Additionally, the strongest edge between outcome expectations and physical activity was found between OE3 and PA1 (0.03), while the strongest edge between body appreciation and physical activity was observed between BA2 and PA1 (0.09). Furthermore, after incorporating health status into the network structure, the results indicate that the revised network comprises 17 nodes and 69 non-zero edges (see Figure 3b and Appendix A, Table A1b). Among the within-variable edges, HS1 and HS2 exhibited the strongest connection (0.76). Regarding inter-variable edges, the strongest connection was found between BA10 and PA1 (0.51). The most robust edge between outcome expectations and body appreciation was observed between OE3 and BA1 (0.33), while the strongest edge between outcome expectations and health status was between OE3 and HS1 (0.06). In the relationship between body appreciation and health status, BA9 and HS2 exhibited the strongest connection (0.10), whereas the strongest edge between physical activity and health status was found between PA2 and HS2 (0.05). Additionally, in the relationship between outcome expectations and physical activity, OE1 and PA1 demonstrated the highest edge strength (0.10).

The network structures at the item level for different health status groups were also analyzed. The results show that the network structures comprise 15 nodes at all levels of health status, with 66 non-zero edges in the high health status group, and 50 non-zero edges in the low health status group (see Figure 3c,d, and Appendix A, Table A1c,d). Among the within-variable edges, the edge between PA1 and PA2 was the strongest in both the high health status group (0.79) and the low health status group (0.64). For inter-variable edges, the strongest connection between outcome expectations and body appreciation in the high health status group was between OE2 and BA1 (0.24), while for the low health status group, it was between OE1 and BA1 (0.13). Regarding the edges between outcome expectations and physical activity, the strongest connection in the high health status group was also between OE2 and BA1 (0.13), whereas there was no connection between outcome expectations and physical activity in the low health status group. For the edges between body appreciation and physical activity, the strongest connection in the high health status group was between BA1 and PA1 (−0.19), while in the low health status group, it was between BA2 and PA2 (0.22). Additionally, regarding the edges between body appreciation and physical activity, the high health status group exhibited 10 connections, whereas the low health status group had only 5 connections.

### 3.5. Centrality Indices

The centrality indices of the networks for the total sample, the high health status group, and the low health status group are presented in Appendix A, Figure A1, Table A2 and Table A3.

In the network comprising outcome expectations, body appreciation, and physical activity for the total sample, BA2 exhibited the highest betweenness and closeness (betweenness: 2.89; closeness: 1.52). OE2 displayed the most substantial strength (1.67), while BA7 had the highest expected influence (1.51). After integrating health status into the network, BA1 showed the highest betweenness and closeness (betweenness: 3.15; closeness: 1.54), whereas BA6 exhibited the greatest strength (1.61) and expected influence (1.56). Additionally, the centrality indices of the network structure for outcome expectations, body appreciation, and physical activity were analyzed across different health status levels. For the high health status group, BA6 had the highest betweenness, and closeness, strength, and expected influence (betweenness: 1.67; closeness: 1.66; strength: 1.57; expected influence: 1.63). However, for the low health status group, BA3 had the highest betweenness and closeness (betweenness: 1.34; closeness:1.56), and BA7 had the highest strength and expected influence (strength: 2.29; expected influence: 1.88).

Moreover, the correlation stability test for centrality indicated that the correlation stability levels of strengths were satisfactory. The correlation stability coefficient for the network structure comprising outcome expectations, body appreciation, and physical activity in the total sample was 0.75. When health status was included, the stability coefficient remained at 0.75. Additionally, the stability coefficient for the high health status group was 0.75, whereas the low health status group exhibited a relatively lower stability coefficient of 0.67. A network comparison test was performed to assess structural differences between high and low health status groups. The results indicate no significant differences in the network structure across the different health status levels (*p* = 0.109). Additionally, there was no significant difference in the overall strength of the network structure between the high and low health status groups (*p* = 0.406). These findings suggest that the network structures constructed for varying health status groups are fundamentally similar.

## 4. Discussion

The present study revealed that outcome expectations positively predicted physical activity and body appreciation played a mediating role in this process. Furthermore, this mediation varied based on individuals’ health status. Specifically, body appreciation promotes physical activity for individuals with a high health status, but the effect is not significant for those with a low health status.

The findings of this study reveal that outcome expectations positively predict physical activity, in line with prior investigations ([29]; [30]; [41]; [51]). This implies that individuals are more inclined to engage in physical activity when they anticipate favorable outcomes from such endeavors. A higher expectation of physical activity outcomes enhances intentions ([33]) and promotes positive beliefs about the behavior ([8]). Moreover, individuals are inclined to perform behaviors in a manner beneficial to achieving the desired outcomes ([28]). Essentially, individuals undertake good health behaviors, including physical activity, to achieve their initial outcome expectations for such behaviors. The validity of these outcome expectations may not be as important as their existence ([28]). Individuals tend to have greater confidence that physical exercise can enhance positive expectations regarding their bodily outcomes rather than negative ones ([29]). Positive health outcome expectations more significantly encourage participation in physical activity compared to negative outcome expectations ([26]). Therefore, it is important to consider that enhancing positive outcome expectations can be effective in promoting intrinsic motivation and lead to more frequent participation in physical activity.

Furthermore, it was found that individuals’ outcome expectations regarding physical activity enhance their favorable perceptions of body image, which in turn fosters their participation in physical activity. This finding is consistent with the HAPA theory, which posits that individuals’ expectations of cultivating a positive self-image, such as body appreciation, are enhanced when they develop expectations concerning health behaviors like physical activity ([47]; [49]). To reduce the disparity between the current outcome expectations and future actual results that may negatively impact their positive body image, individuals may strive to engage in health-promoting behaviors to counteract this discrepancy ([14]). Furthermore, when individuals have respect and appreciation for their bodies, they exhibit a greater tendency to cultivate and attend to their physical health by adopting specific behaviors ([7]). Consequently, physical activity emerges as an effective way to fulfill the physical needs of individuals.

This study further delineated the moderating effect of health status on the relationship between body appreciation and physical activity. Specifically, increased levels of body appreciation positively promoted physical activity participation for individuals with a better health status. Conversely, for individuals with a poor health status, body appreciation exhibited no significant effect on levels of physical activity. Higher levels of body appreciation typically improve health status through health behaviors, thereby cultivating a propensity to advocate for future exercise to maintain congruence between identity and behavior ([13]). Additionally, individuals in robust health may be more likely to experience positive outcomes from physical activity due to their heightened feelings of autonomy and competence, combined with their capacity to derive fulfillment from the challenges and achievements inherent in physical activity ([16]; [45]). In contrast, individuals with poorer health may perceive limitations and challenges when engaging in physical activity due to health constraints, which may reduce their appreciation for such activities ([16]; [45]). Indeed, those who are more inclined to take proactive steps to maintain their health often demonstrate an improved posture and physical condition, as well as a positive attitude toward their bodies ([34]). Individuals in a poorer physical condition may be constrained by their physical limitations, preventing them from engaging in physical exercise and resulting in lower motivation to exercise ([3]; [52]). Therefore, the mediating effect of body appreciation differs for individuals with different levels of health status.

This study analyzed the relationship between the items for the total sample and different health status groups by network analysis. We found that the high health status group had more edges compared to the low health status group, with a greater number of edges linking body appreciation and physical activity in the high health status group than in the low health status group. In the high health status group, the edges connecting item 1 of body appreciation and item 1 of physical activity were the strongest in the relationship between body appreciation and physical activity. This indicated that the role of health status in body appreciation and physical activity may be related to the interconnection of item 1 in both.

Network centrality indices are the foundation for understanding the most decisive nodes in the emerging patterns generated by networks ([44]). Expected influence values highlight which variables are most sensitive to change and thereby realize theoretically desired network configurations through intervention ([35]). The present study demonstrated the strength and expected influence for items 6 and 7 of body appreciation were higher than the other items across different levels of health status, suggesting that they may be the most central features. Therefore, interventions focusing on body appreciation had the potential to lead to improvements across the network.

### Limitations

The present study still had several limitations. Primarily, this study utilized a cross-sectional research design, which remains limited in revealing the dynamic process of modeling relationships over time. Repeated measurements at multiple time points can provide deeper insights into the interactions and trajectories of the variables, which can be considered in future studies. Additionally, the exclusive utilization of samples from the Chinese region entails potential regional bias and expanding the sample size and incorporating diverse sample types in future research endeavors could enhance the robustness, reliability, and applicability of the study’s findings. Thirdly, the self-report method approach may affect the results due to social approvability, etc., and future research could take other approaches to try to control this effect. Finally, although this study utilized traditional fit indices and regression coefficients to evaluate the structural equation model, incorporating effect size calculations could offer deeper insights into the practical significance of the findings ([22]). Future studies should consider integrating effect sizes to improve the interpretability and applicability of the results in practical settings.

## 5. Conclusions

Primarily, outcome expectations positively predicted physical activity. Individuals with optimistic outcome expectations regarding physical activity were more inclined to participate in physical activity. Moreover, body appreciation mediated the relationship between outcome expectations and physical activity, and health status moderated the mediation process of body appreciation. The results indicate that increasing outcome expectations and body appreciation are effective in promoting the frequency of physical activity, and are particularly applicable to healthy people. For those with lower levels of health, the social service system should be improved to provide more personalized support and resources to enhance their health.

## Figures and Tables

**Figure 1 behavsci-15-00394-f001:**
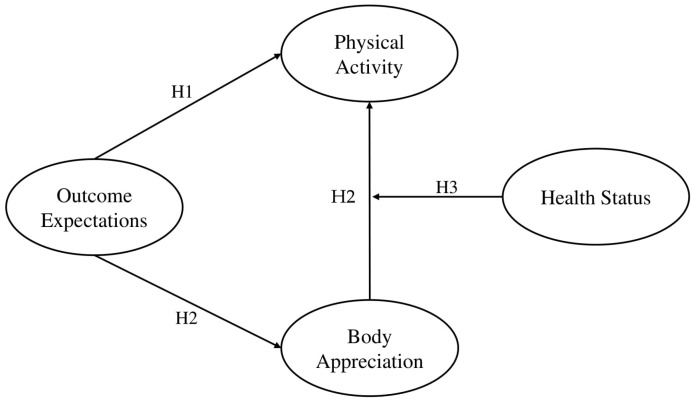
Hypothetical model.

**Figure 2 behavsci-15-00394-f002:**
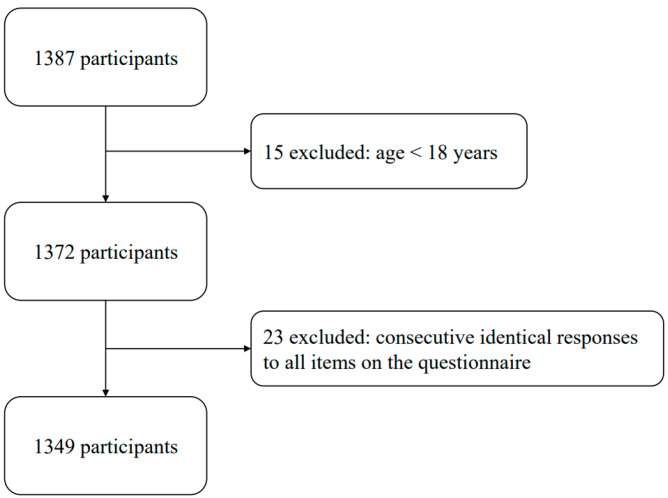
Participant flow.

**Figure 3 behavsci-15-00394-f003:**
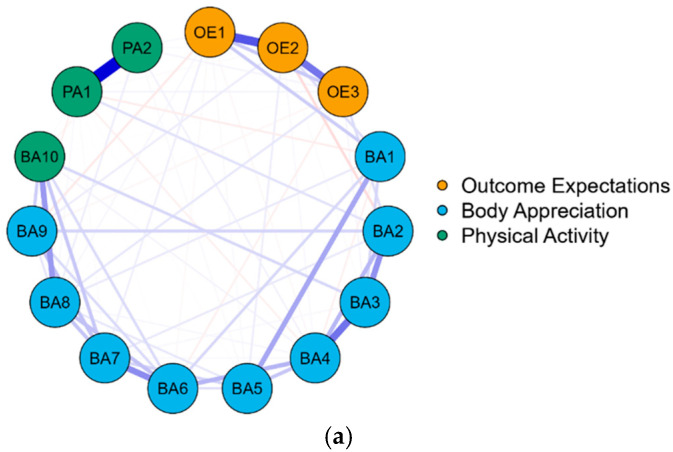
Network structures for total samples and different levels of health status. *Note*: The thickness of the edges represents the strength of the relationship between the two nodes. Edges are blue for positive correlations and red for negative correlations. (**a**) Network structure of outcome expectations, body appreciation, and physical activity for the total sample; (**b**) network structure of outcome expectations, body appreciation, health status, and physical activity for the total sample; (**c**) network structure of outcome expectations, body appreciation, and physical activity for high health status group; (**d**) network structure of outcome expectations, body appreciation, and physical activity for low health status group.

**Table 1 behavsci-15-00394-t001:** Means, standard deviations, and Pearson correlation coefficients of the variables.

	*M*	*SD*	1	2	3	4	5
1. Age	19.42	1.51	-				
2. Subjective Social Class	4.85	1.56	0.02	-			
3. Outcome Expectation	3.67	0.85	0.04	0.14 **	-		
4. Body Appreciation	3.74	0.81	0.01	0.26 ***	0.53 ***	-	
5. Health Status	3.65	0.77	0.01	0.23 ***	0.39 ***	0.56 ***	-
6. Physical Activity	2.80	0.85	0.04	0.09 **	0.20 ***	0.20 ***	0.27 ***

Note: ** *p* < 0.01, and *** *p* < 0.001. The numbers in the first row represent the corresponding variables in the first column (e.g., “1” refers to Age, “2” refers to Subjective Social Class, and so on).

**Table 2 behavsci-15-00394-t002:** The test for the mediating effect.

Effect	Effect Size	SE	LLCI	ULCI	Relative Effect
Total effect	0.18	0.03	0.14	0.23	
Direct effect	0.10	0.03	0.04	0.16	55.56%
Indirect effect	0.08	0.02	0.04	0.16	44.44%

Note: LLCI: lower limit at 95% confidence interval; ULCI: upper limit at 95% confidence interval.

**Table 3 behavsci-15-00394-t003:** The test of the moderated mediation model.

Regression Equation		Regression Coefficient					Overall Fit Index	
**Outcome Variables**	Predictive Variables	*β*	*SE*	LLCI	ULCI	*t*	R^2^	F
BA	OE	0.50	0.03	0.46	0.55	22.147 ***	0.32	157.733
	Gender	0.18	0.05	0.09	0.28	3.751 ***		
	Age	−0.01	0.02	−0.04	0.02	−0.475		
	Subjective Social Class	0.11	0.02	0.08	0.14	7.711 ***		
PA	OE	0.07	0.03	0.02	0.13	2.608 **	0.22	55.249
	BA	0.09	0.03	0.03	0.15	2.789 **		
	HS	0.18	0.03	0.12	0.24	6.082 ***		
	BA × HS	0.06	0.02	0.02	0.10	3.132 **		
	Gender	−0.80	0.05	−0.90	−0.69	−15.059 ***		
	Age	0.03	0.02	−0.01	0.06	1.581		
	Subjective Social Class	0.03	0.02	−0.01	0.06	1.793		

Note: OEs: outcome expectations; BA: body appreciation; HS: health status; PA: physical activity; BA × HS: body appreciation × health status; LLCI: lower limit at 95% confidence interval; ULCI: upper limit at 95% confidence interval; ** *p* < 0.01, and *** *p* < 0.001.

**Table 4 behavsci-15-00394-t004:** Mediating effect values at different levels of health status.

Moderator	Effect Size	SE	LLCI	ULCI
HS (M − 1SD)	0.02	0.02	−0.02	0.06
HS (M)	0.05	0.02	0.01	0.08
HS (M + SD)	0.08	0.02	0.03	0.12

Note: HS: health status; LLCI: lower limit at 95% confidence interval; ULCI: upper limit at 95% confidence interval.

## Data Availability

The raw data supporting the conclusions of this article will be made available by the authors on request.

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
