# Peer review of "Outcome Expectations on Physical Activity: The Roles of Body Appreciation and Health Status"

_behavsci, 2025, doi:10.3390/bs15030394_

Round 1
Reviewer 1 Report
Comments and Suggestions for Authors
To the authors,
Thank you for this interesting manuscript and congratulations! I will first begin by giving a short summary as I understand it with some overall comments. Then, I will provide more detailed comments.
This is a cross-sectional survey study which aimed to examine participants’ expectations of training/exercise outcomes, body appreciation, physical activity involvement, and perceived current health status using self-reported questionnaires. The overall aim was to examine the mechanisms by which these variables influence actual physical activity participation. The authors hypothesize that outcome expectations predict physical activity, while body appreciation mediates the relationship between outcome expectations and physical activity, and perceived health status moderates the relationship between body appreciation and physical activity. Results seem to confirm these hypotheses, and the authors conclude that outcome expectations are important for predicting physical activity, and could be used in the future to individualize exercise programs.
My overall impression of this manuscript is that is well-executed and well-written. A few things I believe would be beneficial are to include a more thorough description of the inclusion and exclusion criteria to be able to understand the population of interest. I also believe that it would be beneficial to discuss the fact that the correlations found are significant, but they are relatively weak, which I would interpret as a need to further explore and take care not to overinterpret the data. Additionally, I believe it would benefit the authors to more thoroughly explain the novelty of this study and the results; as it is written now, it could be interpreted as being a repetition of things already known to the field. For example, the introduction explains that previous studies have shown that outcome expectations predict physical activity, yet the hypothesis is also that outcome expectations will predict physical activity. I understand that the authors are exploring the mediators and moderators, but I believe that this novelty could be made clearer.
Specific comments:
1. Lines 36 and 38: The references provided are quite old. Are there any references that would provide a more up-to-date idea of current physical activity benefits and habits?
2. Line 30: The reference to “Association” seems incomplete? From the reference list, it looks like this is an association that has been categorized as a person. I would recommend looking over the citation guidelines for this type of reference
3. Lines 63-66 and Line 83: On lines 63-66, the authors cite a study which has shown a positive prediction of physical activity, and on line 83, the hypothesis is that this positive prediction will exist. Is this simply a repetition of previous studies? The reader may wonder whether this is a relevant question as it seems it has already been answered. My understanding is that this is a first step – i.e. one must first establish the predictive ability in order to evaluate potential mediators and moderators. If this is a correct interpretation, I would recommend making this clearer for the reader.
4. Line 95: “holds” should be “hold”
5. Lines 162-180 (and Figure 1): I feel that this is more of a methodological description and would fit better in the methods section rather than in the introduction.
6. Section 2.1: The authors reference exclusion of participants, but there is no description of inclusion/exclusion criteria. I would recommend describing which participants are targeted and how they were chosen.
7. Lines 192-193: The terms “invalid samples” and “unreasonable response rate” should be defined so the reader knows what these phrases mean in reference to data collection processes. This is related to my comment 6 regarding exclusion criteria.
8. Also regarding inclusion/exclusion criteria, I believe it would be beneficial to include some form of participant flow chart to see how many people interacted with the survey and different reasons for exclusion (i.e. how many of the 1387 were too young? How many were excluded for any other reason which has not been explained in the text [i.e. exclusion criteria]?).
9. Lines 208-209: To which study are the authors referring? The current study, or the previously referenced study? (It might be helpful to clarify “in the current study” or “in our study”). [This comment is relevant for lines 214, 220, and 226 as well].
10. Lines 222-226: Is this a valid measure of actual physical activity participation?
11. Line 254: “Heal status…” should be “Health status.”
12. Lines 249-255: Are these results from the entire sample? The note in Table 1 seems to indicate subanalyses based on educational level and gender, but I’m not sure I understand where these fit into the results description or the table.
13. Line 264 (Table 1): I would have liked to have in the methods section some explanation of the total scores in order to provide context to the results. For example, the methods describe outcome expectation as being 3 items on a 5-point scale, which basic math tells me would be 15 total; the mean score is 3.67 according to the table. Is this very low or is it in line with other studies?
14. Line 295 [all of section 2.4]: I would recommend including the full names of the variables the first time they are abbreviated to avoid confusion for the reader (I had to go to the figures to double-check what the abbreviations indicated.
15. Line 351: “not significant [for?] those…”
Author Response
To the authors,
Thank you for this interesting manuscript and congratulations! I will first begin by giving a short summary as I understand it with some overall comments. Then, I will provide more detailed comments.
This is a cross-sectional survey study which aimed to examine participants’ expectations of training/exercise outcomes, body appreciation, physical activity involvement, and perceived current health status using self-reported questionnaires. The overall aim was to examine the mechanisms by which these variables influence actual physical activity participation. The authors hypothesize that outcome expectations predict physical activity, while body appreciation mediates the relationship between outcome expectations and physical activity, and perceived health status moderates the relationship between body appreciation and physical activity. Results seem to confirm these hypotheses, and the authors conclude that outcome expectations are important for predicting physical activity, and could be used in the future to individualize exercise programs.
My overall impression of this manuscript is that is well-executed and well-written. A few things I believe would be beneficial are to include a more thorough description of the inclusion and exclusion criteria to be able to understand the population of interest. I also believe that it would be beneficial to discuss the fact that the correlations found are significant, but they are relatively weak, which I would interpret as a need to further explore and take care not to overinterpret the data. Additionally, I believe it would benefit the authors to more thoroughly explain the novelty of this study and the results; as it is written now, it could be interpreted as being a repetition of things already known to the field. For example, the introduction explains that previous studies have shown that outcome expectations predict physical activity, yet the hypothesis is also that outcome expectations will predict physical activity. I understand that the authors are exploring the mediators and moderators, but I believe that this novelty could be made clearer.
[Author Reply]: Thank you very much for your insightful comments and for recognizing the value of our manuscript. We truly appreciate the time and effort you have taken to provide such thoughtful feedback, which has greatly helped us refine and improve our work. We have carefully considered your suggestions and made the necessary revisions accordingly. Below, we provide our point-by-point responses to your comments. Changes made based on your specific suggestions are all marked in yellow-highlighted black text, while modifications incorporating both your feedback and that of another reviewer are marked in yellow-highlighted red text. We sincerely appreciate your constructive input and your support in enhancing the quality of our manuscript.
1. Lines 36 and 38: The references provided are quite old. Are there any references that would provide a more up-to-date idea of current physical activity benefits and habits?
[Author Reply]: Thank you for your suggestion. We have updated the references to reflect more recent research on the benefits of physical activity and current trends. Please see Lines 34–37 for the revised citations.
2. Line 30: The reference to “Association” seems incomplete? From the reference list, it looks like this is an association that has been categorized as a person. I would recommend looking over the citation guidelines for this type of reference.
[Author Reply]: Thank you for your thoughtful feedback. We have reviewed and corrected the citation format to ensure accuracy. The reference is now properly cited as (American College Health Association, 2019) in Line 39.
3. Lines 63-66 and Line 83: On lines 63-66, the authors cite a study which has shown a positive prediction of physical activity, and on line 83, the hypothesis is that this positive prediction will exist. Is this simply a repetition of previous studies? The reader may wonder whether this is a relevant question as it seems it has already been answered. My understanding is that this is a first step – i.e. one must first establish the predictive ability in order to evaluate potential mediators and moderators. If this is a correct interpretation, I would recommend making this clearer for the reader.
[Author Reply]: Thank you for your thoughtful suggestion. The study by Kawaji et al. (2022) investigated 500 patients with pneumoconiosis in Japan and found a significant positive relationship between outcome expectations and physical activity. However, our study mainly focuses on Chinese adolescents (Mage = 19.42, SD = 1.51; 410 males, 939 females). As you rightly pointed out, this is a first step—i.e., one must first establish the predictive ability before evaluating potential mediators and moderators. To make this clearer to readers, we have added the following clarification in Lines 82-85: “Based on this evidence, our study first aims to establish whether outcome expectations positively predict physical activity, which serves as the foundation for exploring potential mediators and moderators in subsequent analyses. Therefore, we propose the following hypothesis:” This should help to address any concerns regarding the repetition of previous studies.
Reference:
Kawaji, T., Hasegawa, T., & Uchiyama, Y. (2022). Dyspnea and outcome expectations are associated with physical activity in persons with pneumoconiosis: a cross-sectional study. BMC Pulmonary Medicine, 22(1), 335. http://doi.org/10.1186/s12890-022-02128-2.
4. Line 95: “holds” should be “hold”.
[Author Reply]: Thank you for your careful review. We have corrected the error, changing “holds” to “hold” in Line 98.
5. Lines 162-180 (and Figure 1): I feel that this is more of a methodological description and would fit better in the methods section rather than in the introduction.
[Author Reply]: Thank you for your valuable feedback. I have revised and optimized this section, and as per your suggestion, moved it to the Data Analysis section in the Methods part. Please refer to Lines 233-235 and Lines 242-250 for the updated content.
6. Section 2.1: The authors reference exclusion of participants, but there is no description of inclusion/exclusion criteria. I would recommend describing which participants are targeted and how they were chosen.
[Author Reply]: Thank you for your valuable suggestion. We have now included a description of the inclusion and exclusion criteria in Section 2.1, “Participants and Research Design”.
The revised text is as follows: “Inclusion criteria were: 1) Age ≥ 18 years; 2) No history of mental illness and normal cognitive ability; 3) No severe physical disabilities or diseases that affect daily activities; 4) Voluntary participation and signed informed consent. Exclusion criteria were: 1) Age < 18 years; 2) History of mental illness or cognitive impairments, such as dementia, etc.; 3) Presence of disability or mobility limitations; 4) Consecutive identical responses to all items on the questionnaire.” This addition can be found in Lines 176-181.
7. Lines 192-193: The terms “invalid samples” and “unreasonable response rate” should be defined so the reader knows what these phrases mean in reference to data collection processes. This is related to my comment 6 regarding exclusion criteria.
[Author Reply]: Thank you for your valuable suggestion. We have added a more detailed description of the inclusion and exclusion criteria, as well as a participant flow chart to provide greater clarity on the reasons for exclusion from the study.
The updated text is as follows: “Inclusion criteria were 1) Age ≥ 18 years; 2) No history of mental illness and normal cognitive ability; 3) No severe physical disabilities or diseases that affect daily activities; 4) Voluntary participation and signed informed consent. Exclusion criteria were: 1) Age < 18 years; 2) History of mental illness or cognitive impairments, such as dementia, etc.; 3) Presence of disability or mobility limitations; 4) Consecutive identical responses to all items on the questionnaire. After excluding 15 participants due to age < 18 years and 23 participants due to consecutive identical responses to all items on the questionnaire, the final valid sample comprised 1349 individuals (Mage = 19.42, SD = 1.51; 939 women), resulting in an effective response rate of 96.97% (Figure 2)”. Please refer to Lines 176-184 and Figure 2 for the updated information.
8. Also regarding inclusion/exclusion criteria, I believe it would be beneficial to include some form of participant flow chart to see how many people interacted with the survey and different reasons for exclusion (i.e. how many of the 1387 were too young? How many were excluded for any other reason which has not been explained in the text [i.e. exclusion criteria]?).
[Author Reply]: Thank you for your valuable suggestion. In response, we have now included a participant flow chart to provide a clear overview of how many individuals interacted with the survey and the reasons for exclusion. Please refer to Figure 2 for the participant flow chart, which details the number of participants excluded based on various criteria, including age and consecutive identical responses.
9. Lines 208-209: To which study are the authors referring? The current study, or the previously referenced study? (It might be helpful to clarify “in the current study” or “in our study”). [This comment is relevant for lines 214, 220, and 226 as well].
[Author Reply]: Thank you for pointing that out. We apologize for the lack of clarity. To address this, we have revised the text to specify “the current study” where needed. The necessary changes have been made in the following lines: Lines 201-202, Line 207, Line 214, and Line 220. We hope this revision enhances the clarity of the text, and we appreciate your understanding.
10. Lines 222-226: Is this a valid measure of actual physical activity participation?
[Author Reply]: Thank you very much for your thoughtful question. I would like to assure you that the measure used in our study to assess physical activity participation is valid. Physical activity was measured using two items derived from a previous study by Schwarzer (2016). This measurement approach has been employed in several prior studies (Baldensperger et al., 2014; Shi et al., 2024; Zhou et al., 2021), including our previous research involving Chinese populations (Shi et al., 2024; Zhou et al., 2021), which further supports its validity in this context. I hope this helps clarify the concern, and I sincerely appreciate your careful review of our work. Thank you again for your valuable feedback.
References:
Schwarzer, R. (2016). Health action process approach (HAPA) as a theoretical framework to understand behavior change. Actualidades en Psicología, 30(121), 119–130. https://doi.org/10.15517/ap.v30i121.23458
Baldensperger, L., Barz, M., Corbett, J., Knoll, N., Lippke, S., & Schwarzer, R. (2014). Physical activity among adults with obesity: Testing the health action process approach. Rehabilitation Psychology, 59. https://doi.org/10.1037/a0035290
Shi, L., Jiang, L., Zhou, S., Zhou, W., & Yang, H. (2024). Self-appreciation is not enough: exercise identity mediates body appreciation and physical activity and the role of perceived stress. Frontiers in Psychology, 15. https://doi.org/10.3389/fpsyg.2024.1377772
Zhou, S., Li, L., Zhao, Y., Cao, Y., Peng, B., and Zheng, L. (2021). Physical activity under stress: a perspective of HAPA and individual differences. International Journal of Environmental Research and Public Health. 18:12144. doi: 10.3390/ijerph182212144
11. Line 254: “Heal status…” should be “Health status.”
[Author Reply]: Thank you very much for your careful review. I apologize for the error. I have corrected “Heal status” to “Health status” in line 262. I appreciate your attention to detail and thank you again for your valuable feedback.
12. Lines 249-255: Are these results from the entire sample? The note in Table 1 seems to indicate subanalyses based on educational level and gender, but I’m not sure I understand where these fit into the results description or the table.
[Author Reply]: [Author Reply]: Thank you very much for your thoughtful comment. Yes, the results presented in Table 1 are based on the entire sample of 1349 participants, after applying the inclusion and exclusion criteria. As for Table 1, it provides descriptive statistics (mean and standard deviation) and Pearson correlation analysis for all variables, including demographic variables (such as gender, age, and subjective social class), outcome expectations, body appreciation, health status, and physical activity. This analysis serves as a preliminary step before conducting structural equation modeling. To enhance the clarity of the table, I have added the following note in Table 1: “The numbers in the first row represent the corresponding variables in the first column (e.g., ‘1’ refers to Gender, ‘2’ refers to Age, and so on); Gender is a binary variable coded as men = 1 and women = 2.”. I hope this improves the readability of the table. Please refer to lines 272-275 for the updated note.
13. Line 264 (Table 1): I would have liked to have in the methods section some explanation of the total scores in order to provide context to the results. For example, the methods describe outcome expectation as being 3 items on a 5-point scale, which basic math tells me would be 15 total; the mean score is 3.67 according to the table. Is this very low or is it in line with other studies?
[Author Reply]: Thank you very much for your valuable suggestion. As you pointed out, the outcome expectation scale consists of 3 items, each rated on a 5-point scale, which gives a possible total score range of 3 to 15. However, I would like to clarify that in the current study, we used the average score, not the total score, for variables such as outcome expectation, body appreciation, health status, and physical activity. For example, for outcome expectation, the average score is calculated as the sum of the three items divided by 3, so the range of the average score is from 1 to 5.
We apologize for not making this clear earlier in the manuscript. Thanks to your helpful comment, we have now added this clarification in the Methods section, where we specify that we used the mean score for each variable. You can find the updated description in Lines 220-221, Lines 206-207, Line 212, and Lines 218-219. We hope this provides a better context for the results. Thank you again for your insightful feedback.
14. Line 295 [all of section 2.4]: I would recommend including the full names of the variables the first time they are abbreviated to avoid confusion for the reader (I had to go to the figures to double-check what the abbreviations indicated.
[Author Reply]: Thank you so much for your valuable suggestion. I completely understand that the use of abbreviations without prior explanation can be confusing for readers. However, as the new abbreviations in Section 3.4 Network Structure are used frequently, providing the full names of the variables could lead to redundancy and excessive text expansion.
To address this issue, I have added an explanation of the abbreviation pattern in Section 2.3 (Data Analysis) to provide clarity for the readers. Specifically, I have included the following description: “In the item-level network analysis, the abbreviation pattern follows the format: [construct abbreviation] + [item number]. For instance, OE1 and OE2 denote the first and second items assessing outcome expectations, respectively. The same convention applies to other constructs.” You can find this explanation in Lines 236-239. I hope this resolves the issue while keeping the content concise. Thank you again for your helpful feedback.
15. Line 351: “not significant [for?] those…”.
[Author Reply]: Thank you for pointing that out. You are absolutely correct, and I apologize for the oversight. I have made the necessary correction to ensure the sentence is clearer. The phrase has now been revised to: “not significant for those...” for better clarity. You can find the updated version in Line 390. Thank you once again for your careful review and valuable feedback.
Reviewer 2 Report
Comments and Suggestions for Authors
On page 5, lines 195–196, the authors should clearly state whether their study received approval from their institutional research board.
Additionally, effect sizes should be provided for all significant findings reported in the manuscript.
More detailed information regarding the participants’ health is needed. For example, are they generally healthy individuals, or do they have any medical conditions? If so, what types of conditions do they have? Do they have psychological disorders or take multiple medications? These are important confounding factors when examining the relationship between health status, physical activity, body appreciation, and outcome expectations. I strongly recommend that the authors include this information in their manuscript and account for these confounding factors in their statistical analysis. It would not be scientifically meaningful to assess the role of health status in the relationship between outcome expectations and physical activity without considering whether participants are healthy or have underlying medical conditions. Additionally, a table summarizing participants’ sociodemographic characteristics and other relevant attributes should be included, as these details are currently missing from the manuscript.
In Table1, the correlation coefficients in columns “1” and “2” are unclear. Can the authors clarify what variables these correlations represent? If column “1” denotes men and column “2” denotes women, the results presented in these columns are confusing.
If Table 2 presents results on the moderating role of health status in the relationship between body appreciation and physical activity, it is unclear why there is an outcome column for body appreciation in this table. The results in Table 2 are confusing, and the authors should provide clarification. Additionally, abbreviation notes should be included for PA.
The analysis for Hypothesis 3, which examines the moderating role of health status in the relationship between body appreciation and physical activity, is not convincing. Simply dichotomizing participants into high- and low-health status groups results in a significant loss of information and oversimplifies the dataset, particularly when health status is assessed via questionnaires, which can be treated as a continuous variable in statistical analysis. As a result, the analysis does not adequately support Hypothesis 3.
Given that numerous studies have highlighted the limitations of dichotomizing participants into groups in the statistical analysis, I strongly suggest re-running the network structure analysis while considering health status as a continuous variable. This approach would provide a more accurate understanding of the connections between outcome expectations, body appreciation, and physical activity. The current statistical methods risk producing misleading inferences and failing to capture the complexity of the dataset, making the results unreliable and unsupportive of the stated hypothesis.
Author Response
Dear Reviewer,
We sincerely appreciate your thoughtful comments and your recognition of our manuscript. Your detailed feedback has been invaluable in helping us refine and enhance our work. We truly value the time and effort you have dedicated to reviewing our study. In response to your suggestions, we have carefully revised the manuscript accordingly. Below, we provide our point-by-point responses to your comments. Revisions made specifically based on your feedback are highlighted in red text, while those reflecting both your input and that of another reviewer are highlighted in yellow-highlighted red text. Thank you again for your constructive insights and for helping us improve the quality of our manuscript.
Best regards,
All authors
1. On page 5, lines 195–196, the authors should clearly state whether their study received approval from their institutional research board.
[Author Reply]: Thank you for your thoughtful suggestion. Prior to submission, we had considered the anonymity and fairness involved in the peer review process, so we only stated that the study had received approval from an institution. Following your recommendation, we have now clearly stated the approval from our institutional research board. Specifically, the previous text “which was approved by [redacted for peer-review] (Approval Number: [redacted for peer-review])” has been revised to “which was approved by the Ethical Committee of Fujian Normal University (20210310), following the Declaration of Helsinki.” Please refer to Lines 186-187 for the update. Thank you again for your valuable feedback.
2. Additionally, effect sizes should be provided for all significant findings reported in the manuscript.
[Author Reply]: Thank you for your thoughtful suggestion. In response, we have incorporated effect size descriptions for the mediation model (see Lines 279–282) and the moderated mediation model (see Lines 306–310) to enhance the clarity and comprehensiveness of our results. Furthermore, we have added Table 2. The test for mediating effect (please see Lines 289–290) and Table 4. Mediating effect values at different levels of health status (please see Lines 311–313) to provide a more detailed presentation of these findings. We sincerely appreciate your valuable feedback, which has helped strengthen the rigor of our analysis.
3. More detailed information regarding the participants’ health is needed. For example, are they generally healthy individuals, or do they have any medical conditions? If so, what types of conditions do they have? Do they have psychological disorders or take multiple medications? These are important confounding factors when examining the relationship between health status, physical activity, body appreciation, and outcome expectations. I strongly recommend that the authors include this information in their manuscript and account for these confounding factors in their statistical analysis. It would not be scientifically meaningful to assess the role of health status in the relationship between outcome expectations and physical activity without considering whether participants are healthy or have underlying medical conditions. Additionally, a table summarizing participants’ sociodemographic characteristics and other relevant attributes should be included, as these details are currently missing from the manuscript.
[Author Reply]: Thank you for your insightful comments and valuable suggestions. We fully understand the importance of considering the participants’ health status in relation to the study’s findings.
In the current study, we specifically focused on generally healthy individuals. To provide more clarity on the participants’ characteristics, we have included the inclusion and exclusion criteria in section 2.1. Participants and Research Design (Lines 176-181) as well as a participant flow chart (Figure 2). The inclusion criteria were as follows: 1) Age ≥ 18 years; 2) No history of mental illness and normal cognitive ability; 3) No severe physical disabilities or diseases that affect daily activities; 4) Voluntary participation with signed informed consent. The exclusion criteria were: 1) Age < 18 years; 2) History of mental illness or cognitive impairments (such as Alzheimer’s disease or dementia); 3) Presence of disability or mobility limitations; 4) Consecutive identical responses to all items on the questionnaire.
Furthermore, we used the self-rated health status measure, which has been widely utilized in previous research and is commonly applied to general populations (Renner & Schwarzer, 2005; Main et al., 2011; Zheng et al., 2020). This measure has been shown to be a valid assessment of health. poor self-rated health status, even among younger individuals (16–24 years old), has been linked to early mortality (Burström & Fredlund, 2001). In studies of elderly Canadians, self-ratings of health were more predictive of 7-year survival than medical records (Mossey & Shapiro, 1982), This may be because self-reporting of perceived health takes into account psychosocial influences on health, such as depression, social isolation, and stress (Kaplan & Camacho, 1983). We hope these clarifications help to address your concern. Thank you once again for your valuable feedback.
References:
Renner, B., & Schwarzer, R. (2005). Risk and health behaviors: documentation of the Scales of the Research Project “Risk Appraisal Consequences in Korea"(RACK). Risk and Health Behaviors: Documentation of the Scales of the Research Project "Risk Appraisal Consequences in Korea” (RACK), 1-55.
Main, A., Zhou, Q., Ma, Y., Luecken, L. J., & Liu, X. (2011). Relations of SARS-Related Stressors and Coping to Chinese College Students' Psychological Adjustment During the 2003 Beijing SARS Epidemic. Journal of Counseling Psychology, 58(3), 410-423. https://doi.org/10.1037/a0023632
Zheng, L., Miao, M., & Gan, Y. Q. (2020). Perceived Control Buffers the Effects of the COVID-19 Pandemic on General Health and Life Satisfaction: The Mediating Role of Psychological Distance. Applied Psychology-Health and Well Being, 12(4), 1095-1114. https://doi.org/10.1111/aphw.12232
Burström, B., & Fredlund, P. (2001). Self-rated health: Is it as good a predictor of subsequent mortality among adults in lower as well as in higher social classes? Journal of Epidemiology and Community Health, 55(11), 836-840. https://doi.org/10.1136/jech.55.11.836
Mossey, J. M., & Shapiro, E. (1982). Self-rated health: a predictor of mortality among the elderly. American Journal of Public Health, 72(8), 800-808. https://doi.org/10.2105/ajph.72.8.800
Kaplan, G. A., & Camacho, T. (1983). Perceived health and mortality: a nine-year follow-up of the human population laboratory cohort. American Journal of Epidemiology, 117(3), 292-304. https://doi.org/10.1093/oxfordjournals.aje.a113541
4. In Table 1, the correlation coefficients in columns “1” and “2” are unclear. Can the authors clarify what variables these correlations represent? If column “1” denotes men and column “2” denotes women, the results presented in these columns are confusing.
[Author Reply]: Thank you for your valuable comment. We recognize the potential ambiguity in Table 1 regarding the correlation coefficients in columns “1” and “2.” To clarify, column “1” corresponds to the variable “Gender” (coded as Men = 1, Women = 2), and column “2” corresponds to “Age.” The correlation coefficients in column “1” indicate associations between gender and other variables. To clarify, we have added the following explanation in the note of Table 1: “The numbers in the first row represent the corresponding variables in the first column (e.g., “1” refers to Gender, “2” refers to Age, and so on); Gender is a binary variable coded as men = 1 and women = 2.” We hope this clarification helps resolve any confusion. Please refer to Lines 272-275 for the updated explanation.
To further enhance clarity and readability, we have replaced the term “Class” with “Subjective Social Class” throughout the manuscript. Additionally, we have updated Section 2.1 Participants and Research Design to explicitly state that demographic information includes age, gender, and subjective social class. We have also clarified that subjective social class was measured using the MacArthur Scale of Subjective Social Status on a 10-point scale (Adler et al., 2000). Please refer to Lines 187-190 for the update.
These revisions ensure that readers can better understand the variables presented in Table 1 and their corresponding measures. We appreciate your suggestion, which has helped improve the clarity and comprehensibility of our manuscript.
Reference:
Adler, N. E., Epel, E. S., Castellazzo, G., & Ickovics, J. R. (2000). Relationship of subjective and objective social status with psychological and physiological functioning: Preliminary data in healthy, white women. Health Psychology, 19(6), 586-592. https://doi.org/10.1037/0278-6133.19.6.586
5. If Table 2 presents results on the moderating role of health status in the relationship between body appreciation and physical activity, it is unclear why there is an outcome column for body appreciation in this table. The results in Table 2 are confusing, and the authors should provide clarification. Additionally, abbreviation notes should be included for PA.
[Author Reply]: Thank you for your valuable feedback. We have revised the previous Table 2, which is now presented as Table 3. The updated table now consists of three sections: Regression equation, Regression coefficient, and Overall fit index, making it more comprehensive and clearer than the previous version. Additionally, to improve readability further, we have updated the table note to explicitly define PA abbreviations. Please refer to Lines 298–301 for details. We appreciate your constructive suggestions, which have significantly improved the clarity and coherence of our results.
6. The analysis for Hypothesis 3, which examines the moderating role of health status in the relationship between body appreciation and physical activity, is not convincing. Simply dichotomizing participants into high- and low-health status groups results in a significant loss of information and oversimplifies the dataset, particularly when health status is assessed via questionnaires, which can be treated as a continuous variable in statistical analysis. As a result, the analysis does not adequately support Hypothesis 3.
[Author Reply]: Thank you very much for your thoughtful feedback. We understand your concern regarding the dichotomization of health status into high and low groups. When conducting the moderated mediation model, we performed a simple slopes analysis to examine the moderating effect. For this analysis, health status was categorized into high (1 SD above the mean) and low (1 SD below the mean) groups, as this approach is commonly used in psychological research and has been widely applied in previous studies (Shi et al., 2024; Yuan et al., 2020; Zhou et al., 2024; Zhou et al., 2021). For example, Shi et al. (2024) employed a similar strategy when exploring the moderating role of perceived stress in the relationship between body appreciation and exercise identity, and Yuan et al. (2020) used this approach in their study on the moderating role of physical activity in the relationship between mental health and quality of life. Thus, we believe that this method of analysis is valid and supports our hypothesis on the moderating effect.
Additionally, to ensure the appropriateness of this approach, we have reconstructed and re-examined the structural equation model using SPSS PROCESS and updated the results section. The findings remain consistent with the previous analysis. We have also added a description of the use of SPSS in Section 2.3 Data Analysis (please see Line 222 and Lines 225-227) and updated the SEM results accordingly (please refer to Lines 279-288, Line 295, and Lines 303-310).
We appreciate your comments and hope this clarifies our approach. Please let us know if you have further questions or suggestions.
References:
Shi, L., Jiang, L., Zhou, S., Zhou, W., & Yang, H. (2024). Self-appreciation is not enough: exercise identity mediates body appreciation and physical activity and the role of perceived stress. Frontiers in Psychology, 15. https://doi.org/10.3389/fpsyg.2024.1377772
Yuan, Y., Li, J., Jing, Z., Yu, C., Zhao, D., Hao, W., & Zhou, C. (2020). The role of mental health and physical activity in the association between sleep quality and quality of life among rural elderly in China: A moderated mediation model. Journal of Affective Disorders, 273, 462-467. https://doi.org/https://doi.org/10.1016/j.jad.2020.05.093
Zhou, S., Leng, M., Zhang, J., Zhou, W., Lian, J., & Yang, H. (2024). Parental emotional warmth and adolescent internet altruism behavior: a moderated mediation model. Humanities and Social Sciences Communications, 11(1), 446. https://doi.org/10.1057/s41599-024-02870-4
Zhou, S., Li, L., Zhao, Y., Cao, Y., Peng, B., and Zheng, L. (2021). Physical activity under stress: a perspective of HAPA and individual differences. International Journal of Environmental Research and Public Health. 18:12144. https://doi.org/10.3390/ijerph182212144
7. Given that numerous studies have highlighted the limitations of dichotomizing participants into groups in the statistical analysis, I strongly suggest re-running the network structure analysis while considering health status as a continuous variable. This approach would provide a more accurate understanding of the connections between outcome expectations, body appreciation, and physical activity. The current statistical methods risk producing misleading inferences and failing to capture the complexity of the dataset, making the results unreliable and unsupportive of the stated hypothesis.
[Author Reply]: Thank you for your insightful suggestion. We appreciate your concern regarding the limitations of dichotomizing health status in statistical analysis. As network analysis provides a unique perspective by examining item-level relationships and centrality indices, it serves as a valuable complement to structural equation modeling (SEM), which assesses relationships at the variable level. The network structure analysis allows us to explore how the relationships among outcome expectations, body appreciation, and physical activity change across different levels of the moderator, thereby enhancing the interpretability of our findings.
In response to your recommendation, we have re-run the network analysis by incorporating health status as a continuous variable. The added results, including network structure, centrality indices, and stability of edge weights, have been integrated into the manuscript. Specifically, the new network structure consists of 17 nodes and 69 non-zero edges, with key connections detailed in Lines 323–334. Additionally, the centrality indices are reported in Lines 356–367, and the stability analysis results are provided in Lines 372–378. Furthermore, we have included the corresponding figures and tables in the manuscript, including Figure 3b, Appendix Figure 1b, Appendix Figure 2b, Appendix Table 1b, and Appendix Table 3. We sincerely appreciate your valuable feedback, which has helped strengthen the methodological rigor of our study and enhance the clarity of our results.
Round 2
Reviewer 1 Report
Comments and Suggestions for Authors
To the authors:
Thank you very much for your thorough and thoughtful responses! I believe all of my questions and concerns have been addressed and the quality of the manuscript has been greatly improved!
Author Response
To the authors:
Thank you very much for your thorough and thoughtful responses! I believe all of my questions and concerns have been addressed and the quality of the manuscript has been greatly improved!
[Author Reply]: Thank you very much for your positive feedback and for taking the time to review our manuscript. We sincerely appreciate your insightful comments and suggestions, which have significantly contributed to improving the quality of our work.
Reviewer 2 Report
Comments and Suggestions for Authors
- The authors should include an effect size analysis in the Data Analysis section and explain how to interpret the strength of the effect sizes in the Data Analysis and result sections. This will provide a better understanding of the practical significance of the findings.
- In Table 1, it is unclear how Pearson correlation was applied between a binary variable (e.g., gender) and continuous variables. Pearson correlation is designed to measure the linear relationship between two continuous variables, making its use with binary data statistically questionable. The authors should provide correlation figures along with an explanation to justify this analysis or consider alternative statistical approaches.
- In Table 3, the relationship between Body Appreciation (BA) and the predictor variables (Outcome Expectation, Gender, Age, Subjective Social Class) does not clearly contribute to understanding the moderating role of Health Status (HS) in the relationship between BA and Physical Activity. Since in the second model, the interaction between BA and HS already addresses the main research question, the authors should clarify the purpose and relevance of these additional analyses. Additionally, while the authors argue that many psychological studies dichotomize variables to examine moderation effects, prior research has highlighted significant biases and information loss associated with this approach. Given these concerns, I strongly recommend treating Health Status as a continuous variable in the model rather than dichotomizing it. Relying on a method simply because it has been used in previous studies is not a strong justification, especially when evidence suggests it can be misleading. Reanalyzing the data with Health Status as a continuous variable would provide a more robust and statistically meaningful interpretation of the findings.
Author Response
1. The authors should include an effect size analysis in the Data Analysis section and explain how to interpret the strength of the effect sizes in the Data Analysis and result sections. This will provide a better understanding of the practical significance of the findings.
[Author Reply]: Thank you very much for your valuable suggestion. We reviewed relevant literature and found that different studies have varying opinions on calculating the effect size for Structural Equation Modeling (SEM). For instance, Flora et al. (2025) suggested that the effect size for SEM can be represented by unstandardized coefficients, standardized coefficients, and R², which have already been reported in our study. Additionally, Gomer et al. (2019) proposed a new indicator of effect size (i.e., ε). However, this approach has not been widely adopted, and its validity has yet to gain broad recognition. Due to the complexity of the calculation of ε, we currently do not have a full understanding of the method. Therefore, we have acknowledged this limitation in the study, specifically in Lines 470-474 (red text): “Finally, although this study utilized traditional fit indices and regression coefficients to evaluate the structural equation model, incorporating effect size calculations could offer deeper insights into the practical significance of the findings (Gomer et al., 2019). Future studies should consider integrating effect sizes to improve the interpretability and applicability of the results in practical settings.”
Furthermore, if it is convenient for you, we would greatly appreciate your guidance on how to calculate the effect size for SEM. If you could recommend any relevant articles or tutorials, we would be sincerely grateful and eager to learn. If we are able to master this method in the future, we will include the effect size calculation in our subsequent revision and research. Thank you again for your insightful suggestions and guidance.
References:
Flora, D. B., Crone, G., & Bell, S. M. (2025). Effect Size Interpretation in Structural Equation Models. Structural Equation Modeling: A Multidisciplinary Journal, 1-8. http://doi.org/10.1080/10705511.2025.2459768.
Gomer, B., Jiang, G., & Yuan, K.-H. (2019). New Effect Size Measures for Structural Equation Modeling. Structural Equation Modeling: A Multidisciplinary Journal, 26(3), 371-389. http://doi.org/10.1080/10705511.2018.1545231.
2. In Table 1, it is unclear how Pearson correlation was applied between a binary variable (e.g., gender) and continuous variables. Pearson correlation is designed to measure the linear relationship between two continuous variables, making its use with binary data statistically questionable. The authors should provide correlation figures along with an explanation to justify this analysis or consider alternative statistical approaches.
[Author Reply]: Thank you for your valuable suggestion. We sincerely appreciate your insightful feedback regarding the application of Pearson correlation. In response to your comment, we have removed the Pearson correlation results between the binary variable (i.e., gender) and other variables from Table 1. Please refer to lines 273-275 (red text) for the updated version. Thank you again for your careful review and thoughtful feedback.
3. In Table 3, the relationship between Body Appreciation (BA) and the predictor variables (Outcome Expectation, Gender, Age, Subjective Social Class) does not clearly contribute to understanding the moderating role of Health Status (HS) in the relationship between BA and Physical Activity. Since in the second model, the interaction between BA and HS already addresses the main research question, the authors should clarify the purpose and relevance of these additional analyses. Additionally, while the authors argue that many psychological studies dichotomize variables to examine moderation effects, prior research has highlighted significant biases and information loss associated with this approach. Given these concerns, I strongly recommend treating Health Status as a continuous variable in the model rather than dichotomizing it. Relying on a method simply because it has been used in previous studies is not a strong justification, especially when evidence suggests it can be misleading. Reanalyzing the data with Health Status as a continuous variable would provide a more robust and statistically meaningful interpretation of the findings.
[Author Reply]: Thank you very much for your insightful comments and suggestions. Regarding Table 3, it presents the results of a single SEM model (i.e., Figure 1. Hypothetical Model), rather than results from two separate SEM models. In fact, Table 3 reflects the results from Figure 1 after controlling for gender, age, and subjective social class. If represented visually, Table 3 would correspond to Letter Figure 1. While the lower part of Table 3, which reports the interaction between Body Appreciation (BA) and Health Status (HS), already demonstrates the moderating role of HS in the relationship between BA and Physical Activity, omitting the other results could lead to an incomplete and potentially misleading interpretation. Specifically, the analytical approach of this study follows a stepwise process, where a mediation model was first established, and then a moderated mediation model was constructed based on it. If we were to remove the upper part of Table 3 and retain only the interaction between BA and HS, it might give the impression that we conducted a simple moderation model (as illustrated in Letter Figure 2), rather than the intended moderated mediation model (Letter Figure 1). For this reason, we believe it is important to retain the upper portion of Table 3 to emphasize that our study examines a moderated mediation model. However, to enhance clarity and avoid potential confusion, we have revised the title of Table 3 to “The test of the moderated mediation model” (Line 298, red text), making it clearer to readers that the results pertain to a moderated mediation analysis rather than a simple moderation model.
Letter Figure 1. Moderated mediation model
Letter Figure 2. Simple moderated model
Regarding the moderation analysis, we would like to clarify that HS was treated as a continuous variable when constructing the moderated mediation model, rather than being dichotomized. If the moderation effect is significant, Simple Slope Analysis is commonly conducted to further interpret the interaction effect. In this analysis, the moderator (HS) is categorized into high (M+1SD) and low (M-1SD) groups based on its mean and standard deviation. This approach has been widely adopted in numerous studies, including those published in high-impact journals, such as Nature Human Behaviour (Dick et al., 2021), Nature Mental Health (Chan et al., 2024), Nature Communications (Speer et al., 2024), and Development Psychology (Guimond et al., 2018; Llewellyn & Rudolph, 2014; Mabbe et al., 2019; Scrimgeour et al., 2016; Zhang et al., 2023), as well as recently published articles in Behavioral Sciences (Belgasm et al., 2025; Chang et al., 2025; Guo et al., 2025; Li et al., 2025; Liu & Li, 2025; Ren et al., 2025; Song & Chang, 2025; Trombetta et al., 2025; Wang et al., 2024; Wang & Hou, 2025). The broad acceptance of this method underscores its validity.
Despite our best efforts, we have yet to identify alternative approaches to conducting Simple Slope Analysis. We would be truly grateful if you could share any relevant resources or recommend alternative methods. Your expertise and guidance would be invaluable to us, and we sincerely appreciate your thoughtful feedback.
References:
Belgasm, H., Alzubi, A., Iyiola, K., & Khadem, A. (2025). Interpersonal Conflict and Employee Behavior in the Public Sector: Investigating the Role of Workplace Ostracism and Supervisors' Active Empathic Listening. Behavioral Sciences, 15(2), Article 194. http://doi.org/10.3390/bs15020194.
Chan, S. Y., Ngoh, Z. M., Ong, Z. Y., Teh, A., Kee, M. Z. L., Zhou, J. H., Fortier, M. V., Yap, F., Macisaac, J. L., Kobor, M. S., Silveira, P. P., Meaney, M. J., & Tan, A. P. (2024). The influence of early-life adversity on the coupling of structural and functional brain connectivity across childhood. Nature Mental Health, 2(1). http://doi.org/10.1038/s44220-023-00162-5.
Chang, Y. J., Martirosyan, A., Lim, H. W., & Yoo, J. W. (2025). Effects of Failure Acceptance, Entrepreneurial Orientation, and Social Safety Net on Entrepreneurial Intention: A Moderated Mediation Analysis of Korean Employees. Behavioral Sciences, 15(1), Article 28. http://doi.org/10.3390/bs15010028.
Dick, A. S., Silva, K., Gonzalez, R., Sutherland, M. T., Laird, A. R., Thompson, W. K., Tapert, S. F., Squeglia, L. M., Gray, K. M., Nixon, S. J., Cottler, L. B., La Greca, A. M., Gurwitch, R. H., & Comer, J. S. (2021). Neural vulnerability and hurricane-related media are associated with post-traumatic stress in youth. Nature Human Behaviour, 5(11), 1578-+. http://doi.org/10.1038/s41562-021-01216-3.
Guimond, F. A., Brendgen, M., Correia, S., Turgeon, L., & Vitaro, F. (2018). The Moderating Role of Peer Norms in the Associations of Social Withdrawal and Aggression With Peer Victimization. Developmental Psychology, 54(8), 1519-1527. http://doi.org/10.1037/dev0000539.
Guo, Z. R., Xu, T., & Li, H. J. (2025). The Moderating Effect of Self-Construal on the Relationship Between Mindfulness and Forgiveness. Behavioral Sciences, 15(2), Article 195. http://doi.org/10.3390/bs15020195.
Li, X. P., Xu, C. L., Chen, W. Y., & Tian, J. (2025). Physical Exercise and Sleep Quality Among Chinese College Students: The Mediating Role of Self-Control and the Moderating Role of Mindfulness. Behavioral Sciences, 15(2), Article 232. http://doi.org/10.3390/bs15020232.
Liu, X., & Li, Y. X. (2025). Examining the Double-Edged Sword Effect of AI Usage on Work Engagement: The Moderating Role of Core Task Characteristics Substitution. Behavioral Sciences, 15(2), Article 206. http://doi.org/10.3390/bs15020206.
Llewellyn, N., & Rudolph, K. D. (2014). Individual and Sex Differences in the Consequences of Victimization: Moderation by Approach and Avoidance Motivation. Developmental Psychology, 50(9), 2210-2220. http://doi.org/10.1037/a0037353.
Mabbe, E., Vansteenkiste, M., Brenning, K., De Pauw, S., Beyers, W., & Soenens, B. (2019). The Moderating Role of Adolescent Personality in Associations Between Psychologically Controlling Parenting and Problem Behaviors: A Longitudinal Examination at the Level of Within-Person Change. Developmental Psychology, 55(12), 2665-2677. http://doi.org/10.1037/dev0000802.
Ren, X. X., Xu, H., Yue, T., & Wang, T. (2025). The Effects of Value Conflicts on Stress in Chinese College Students: A Moderated Mediation Model. Behavioral Sciences, 15(2), Article 104. http://doi.org/10.3390/bs15020104.
Scrimgeour, M. B., Davis, E. L., & Buss, K. A. (2016). You Get What You Get and You Don't Throw a Fit!: Emotion Socialization and Child Physiology Jointly Predict Early Prosocial Development. Developmental Psychology, 52(1), 102-116. http://doi.org/10.1037/dev0000071.
Song, S., & Chang, P. C. (2025). The Impact of Benevolent Sexism on Women's Career Growth: A Moderated Serial Mediation Model. Behavioral Sciences, 15(1), Article 59. http://doi.org/10.3390/bs15010059.
Speer, S. P. H., Mwilambwe-Tshilobo, L., Tsoi, L., Burns, S. M., Falk, E. B., & Tamir, D. I. (2024). Hyperscanning shows friends explore and strangers converge in conversation. Nature Communications, 15(1), Article 7781. http://doi.org/10.1038/s41467-024-51990-7.
Trombetta, T., Fusco, C., Rollè, L., & Santona, A. (2025). Untangling Relational Ties: How Internalized Homonegativity and Adult Attachment Shape Relationship Quality in Lesbian and Gay Couples. Behavioral Sciences, 15(2), Article 205. http://doi.org/10.3390/bs15020205.
Wang, H., Yue, T., & Luo, H. J. (2024). The Impact of Self-Transcendence on Anxiety Among Chinese College Students: The Moderating Roles of Self-Enhancement and Dominant Self-Construal. Behavioral Sciences, 14(11), Article 1105. http://doi.org/10.3390/bs14111105.
Wang, P., & Hou, Y. (2025). The Effects of Hotel Employees' Attitude Toward the Use of AI on Customer Orientation: The Role of Usage Attitudes and Proactive Personality. Behavioral Sciences, 15(2), Article 127. http://doi.org/10.3390/bs15020127.
Zhang, J. L., Deng, H. M., Liu, T. T., & Mu, S. K. (2023). Self-experience consistency and life satisfaction: The mediating role of the need for relatedness and the moderating role of Zhong-yong thinking. Humanities & Social Sciences Communications, 10(1), Article 334. http://doi.org/10.1057/s41599-023-01846-0.
